# The Influence of Laser Process Parameters on the Adhesion Strength between Electroless Copper and Carbon Fiber Composites Determined Using Response Surface Methodology

**DOI:** 10.3390/mi14122168

**Published:** 2023-11-29

**Authors:** Xizhao Wang, Jianguo Liu, Haixing Liu, Zhicheng Zhou, Zhongli Qin, Jiawen Cao

**Affiliations:** 1Institute of Laser and Intelligent Manufacturing Technology, South-Central Minzu University, Wuhan 430074, China; wangxizhao@scuec.edu.cn (X.W.); liuhaixing98@gmail.com (H.L.); 2Wuhan National Laboratory for Optoelectronics (WNLO), Huazhong University of Science and Technology (HUST), Wuhan 430074, China; liujg@mail.hust.edu.cn (J.L.); m202170333@hust.edu.cn (Z.Z.); 3School of Electronics and Information Engineering, Hubei University of Science and Technology, Xianning 437100, China; caojiawen@hbust.edu.cn

**Keywords:** carbon fiber composites, adhesion strength, response surface methodology, optimization, laser process parameters

## Abstract

Laser process technology provides a feasible method for directly manufacturing surface-metallized carbon fiber composites (CFCs); however, the laser’s process parameters strongly influence on the adhesion strength between electroless copper and CFCs. Here, a nanosecond ultraviolet laser was used to fabricate electroless copper on the surface of CFCs. In order to achieve good adhesion strength, four key process parameters, namely, the laser power, scanning line interval, scanning speed, and pulse frequency, were optimized experimentally using response surface methodology, and a central composite design was utilized to design the experiments. An analysis of variance was conducted to evaluate the adequacy and significance of the developed regression model. Also, the effect of the process parameters on the adhesion strength was determined. The numerical analysis indicated that the optimized laser power, scanning line interval, scanning speed, and pulse frequency were 5.5 W, 48.2 μm, 834.0 mm/s, and 69.5 kHz, respectively. A validation test confirmed that the predicted results were consistent with the actual values; thus, the developed mathematical model can adequately predict responses within the limits of the laser process parameters being used.

## 1. Introduction

As appealing materials, carbon fiber composites (CFCs) are extensively applied in many fields, including the automobile and aircraft/aerospace industries, due to their unique properties: excellent mechanical performance, low density, high thermal stability, and so on [1,2,3]. However, they also have some obvious disadvantages, such as poor electrical conductivity and low erosion resistance, which limit their further application in the field of aircraft structure/manufacture, as materials with better electrical conductivity can effectively avoid damage from lightning strikes [4]. Therefore, there is an urgent need to establish a project that focuses on protecting the composite outer skin of aircraft against lightning damage. The protective layer formed via the metallization of CFC surfaces exhibits excellent electrical conductivity, making it one of the most effective solutions to solve this problem.

In recent years, there has been growing interest in the surface metallization of CFCs [5]. To make CFCs electrically conductive, metallic materials can be coated onto the polymer surface [6]. However, the intrinsic hydrophobicity of CFC surfaces leads to poor adhesion between the metal layer and CFCs, which severely limits their wide application. Therefore, obtaining a high-strength metal layer has become critical in the fabrication of conductive CFCs. Various technologies have been developed to fabricate metal layers with high adhesion to CFC surfaces, including electroless plating, electrodeposition, spraying, and plasma etching [7,8,9,10]. For instance, Chen et al. deposited copper onto CFCs using an electroless method and obtained copper-coated CFCs with good adhesion and excellent conductivity [11]. Wang et al. reported a pulse-reverse electrodeposition process to fabricate Ni-coated CFCs, which exhibited good adhesion [12]. Archambault et al. successfully cold-sprayed copper onto CFCs to prepare Cu-coated CFCs with an adhesion strength of 2.6 MPa ± 0.8 MPa [13]. Prysiazhnyi et al. employed nitrogen plasma modification for surface treatment to enhance the adhesion between electroless copper and CFCs [14]. However, the above-mentioned approaches suffer from inherent disadvantages, e.g., a tedious operation process, high cost, corrosive or poisonous chemical reagents, poor adhesion strength, and so on, which have seriously limit their practical applications. 

In comparison, laser process technology offers numerous many advantages, including noncontact operation, high precision, region selection, controllability, simplicity, and applicability to a wide range of materials [15,16,17,18,19], which make it a promising method for preparing metal-coated CFCs with high adhesion. For instance, Gustke et al. proposed a pulsed laser roughening method for enhancing the adhesion strength between a sprayed copper layer and CFCs. It was shown that the adhesion strength was increased by 200% [20]. Li et al. investigated the effects of the pretreatment of a picosecond infrared laser and an excimer ultraviolet laser on improving the shear strength between an aluminum alloy and CFCs, respectively. It was revealed that the shear strength was determined based on chemical bonding rather than mechanical interlocking [21]. Palavra et al. studied the effect of laser surface pretreatment parameters on the adhesion strength between a titanium layer and CFCs, including the laser power and pulse energy [22]. In summary, most researchers in this field have primarily focused on the impact of laser treatment on enhancing the adhesion strength between metal layers and CFCs. The influence of the key input parameters of the laser process, such as the laser power, scanning line interval, scanning speed, and pulse frequency, on the adhesion strength between metal layers and CFCs have not been taken into account. In addition, the interactive effects of process input parameters on adhesion strength have not been studied.

To understand their effects on adhesion strength, laser process parameters should be extensively analyzed and optimized. Response surface methodology (RSM), used for experimental design, is among the most effective optimization techniques [23], which can not only reveal the relationship between the input variables and output responses but also predict the optimal output responses through numerical optimization under certain conditions. And, the RSM is often employed in numerous fields to find the optimal parametric combination from a group of given variables for achieving the desired output responses [24,25,26]. 

Generally, the adhesion strength between electroless copper and CFCs is influenced by numerous factors. Thus, to obtain excellent adhesion strength with copper plating, it is necessary to investigate the effect of different laser process parameters on the adhesion strength. However, to the best of our knowledge, the influence of the key laser process parameters on the adhesion strength between electroless copper and CFCs is relatively poorly understood. The interactive effects of laser process parameters on adhesion strength have not been reported.

Central composite design (CCD) is the primary type of RSM, and it is effective for obtaining more information about experimental variables and experimental errors with the fewest experimental cycles. In this study, a standard RSM with a CCD was applied to develop a model for optimizing the laser process parameters to achieve good adhesion strength. The key laser process parameters considered were laser power, scanning line interval, scanning speed, and pulse frequency. An analysis of variance (ANOVA), main effect plots, contour plots, and the corresponding 3D response surface were employed to assess the effect of each factor on the adhesion strength between electroless copper and CFCs. Furthermore, a mathematical model was developed to predict the optimal adhesion strength.

## 2. Materials and Methods

### 2.1. Materials

Commercial CFC sheets, which had a thickness of 2.0 mm and consisted of carbon fiber and epoxy polymer, were purchased from Beijing Plastic Manufacturing Co., Ltd. (Beijing, China). Before laser ablation, the CFC sheet was cut into small pieces (30.0 mm × 30.0 mm). Next, the samples were rinsed ultrasonically with acetone for 15 min, followed by rinsing with deionized water for an additional 15 min. Pentahydrate copper sulfate (CuSO_4_·5H_2_O) and EDTA disodium (Na_2_EDTA) were purchased from Sinopharm Chemical Reagent Co., Ltd. (Shanghai, China). Formaldehyde (HCHO), sodium hydroxide (NaOH), and potassium sodium tartrate tetrahydrate (C_4_H_12_KNaO_10_) were purchased from Aladdin Reagent (Minneapolis, Michigan, USA). All chemicals were analytical grade and used directly without further purification. 

### 2.2. Fabrication of Electroless Copper on CFCs

Figure 1 illustrates the fabrication process of electroless copper on the CFC surface, including formation, coarsening, removal and electroless plating: (1) The CFC sheet could not initiate the process of electroless plating itself due to the lack of catalytically active centers (i.e., active seeds). Many precious metals, such as gold (Au), palladium (Pd), and silver (Ag), are usually used as active seeds. Among them, Pd is optimal due to its excellent catalytic activity. As shown in Figure 1, the CFC sheets were first immersed in a 1.0 g/L PdCl_2_ aqueous solution for 10 min and then taken out and dried. Thus, the CFC sheets coated with PdCl_2_ films were obtained. (2) The coated CFC sheets were coarsened using laser direct ablation equipment (Figure 2a). As shown in Figure 2b, the direct laser ablation system was mainly composed of a 355 nm nanosecond laser, an optical system, a multiaxis workbench system, and a controlling system. The maximum average laser power was about 20 W, the highest scanning speed was about 10,000 mm/s, the pulse width was about 16 ns, the value of the beam quality factor *M*^2^ was less than 1.2, and the repetition rate ranged from 200 kHz to 2 MHz. A two-mirror galvanometric scanner with an F-theta objective lens was employed to focus and scan the laser beam in the x−y direction. The Gaussian-profile laser beam at 1/e^2^ of its maximum intensity had a focused spot diameter of about 15 μm. During laser direct ablation, the CFC sheet (Figure 3a)-coated PdCl_2_ films were mounted on an x-y-z translation stage precisely controlled using a computer. A line-by-line scanning method in the x direction and then in the y direction was used to fabricate a rough surface (Figure 3b–f) on the CFC sheets for obtaining the electroless copper with high adhesion in an atmospheric environment. (3) The ablated CFC sheets were cleaned ultrasonically in deionized water to selectively remove the active seeds (i.e., PdCl_2_) in the non-laser-irradiated zone, while keeping the active seeds in the laser-irradiated zone. (4) The cleaned CFC sheets were immersed into a commercial copper bath solution for selective copper plating. The obtained electroless coppers on the surfaces of CFC sheet are shown in Figure 4.

### 2.3. Measurements and Characterization

The morphological structures of the laser-ablated CFC surfaces were analyzed using a Nova NanoSEM 450 scanning electron microscope (SEM, FEI, Hillsboro, OR, USA). The adhesion strength between the prepared copper plating layer and CFC sheet was measured via the vertical pulling force method shown in Figure 5. First, a copper plating layer on the CFC surface with dimensions of 2.0 mm × 2.0 mm was prepared. Then, it was welded to a metal wire using lead–tin solder. Next, the pulling force of the dynamometer was gradually increased along the direction perpendicular to the CFC sheet, and the maximum tension pulling force (*F_M_*) displayed on the dynamometer was recorded until the copper metal layer was pulled off. The adhesion strength (*Y_M_*) of the copper plating layer on the CFC surface was calculated according to Equation (1) [27].
(1)$$Y_{M}=\frac{F_{M}}{S}$$
where *S* is the surface area of the electroless copper. For each sample, five parallel tests of the adhesion strength *Y_M_* were performed to obtain an average value.

### 2.4. Modeling of Pulsed Laser Process

#### 2.4.1. Response Surface Methodology

RSM is a collection of mathematical and statistical techniques, which is usually used to model and explore a problem in which a response is influenced by multiple variables. The RSM can predict the relationship between output responses and input independent variables in a specific range, including optimization methods to explore the optimum values of the input independent parameters that produce the desirable output responses [28]. Based on measurable, continuous, and controllable input independent variables (*x*_1_, *x*_2_, *x*_3_, …, *x*_k_), and when the error is negligible, the linear output response *Y_M_* can be described as follows:(2)$$Y_{M}=\varphi \left(x_{1},x_{2},x_{3},\ldots ,x_{k}\right)+\varepsilon$$
where *φ* represents the true response function, the accurate form of which is unknown and very complex; *ε* is a term associated with other sources of variability not taken into account in *φ*. Commonly, *ε* represents the effects of measurement error on response and background noise, as well as the influence of other variables, and so on. In general, *ε* is often applied as a statistical error, which is assumed to have normal distribution with a mean value of zero and a variance value of σ^2^. 

Generally, the second-order polynomial regression Equation (3) is used in RSM.
(3)$$Y_{M}=\beta_{0}+\sum_{i=1}^{m}\beta_{i}x_{i}+\sum_{i=1}^{m}\sum_{j=1}^{m}\beta_{ij}x_{i}x_{j}+\sum_{i=1}^{m}\beta_{ii}x_{i}^{2}+\varepsilon$$
where *x_i_* and *x_j_* are the coded parameter variables; *β*_0_ is a constant; *β_i_*, *β_ij_,* and *β_ii_* represent the coefficients of the linear, quadratic, and interactive effects, respectively. These coefficients can be obtained through the fitting of experimental data [29]. The fitting precision of the mathematical model was evaluated using, the coefficient of determination *R*^2^, adjusted *R*^2^, predicted *R*^2^, and adequate precision. The *F*-test and *p*-value were applied to check the significance of the regression coefficients.

#### 2.4.2. Experimental Design

Based on previous single-factor experiments, the laser power (*A*), scanning line interval (*B*), scanning speed (*C*), and laser pulse frequency (*D*), were chosen as parameter variables and the adhesion strength as the response variable. A standard RSM design with CCD was applied to evaluate the effects of the laser process parameter variables (*A*, *B*, *C*, and *D*) on the response variable (*Y_M_*). Incorporating the five coded levels (−2, −1, 0, 1, 2) for each of the four variables resulted in a total of 30 experiments. The laser process parameters and their levels investigated in this study are shown in Table 1.

The statistical software Design-Expert V11.0 was employed to establish the correlations between the variation in the laser process parameters and the adhesion strength (*Y_M_*). The results obtained from experiments are listed in Table 2.

## 3. Results and Discussion

### 3.1. Analysis of Variance

The analysis of variance is a well-known and powerful statistical analysis method [30]. In this study, ANOVA was applied to test whether the laser process parameters had a significant effect on the adhesion strength to investigate the prediction ability of the developed regression models in the design space. The ANOVA table for the adhesion strength (*Y_M_*) of electroless copper on CFCs is shown in Table 3. 

The associated *p*-value for the model was less than 0.05, indicating that the model term was statistically significant [31]. In Table 3, the ANOVA results show that the laser power (*A*), scanning speed (*C*), the quadratic effect of the laser power(*A*^2^), scanning line interval (*B*^2^), scanning speed (*C*^2^), and laser pulse frequency (*D*^2^), along with the interaction effect of laser power and laser pulse frequency (*AD*), were the significant model terms associated with adhesion strength. The other terms of the mathematical model were not significant, and they needed to be eliminated through a backward elimination process to improve the adequacy of the model.

The ANOVA table for the simplified quadratic model is shown in Table 4. The fitness of the adhesion strength model *R*^2^ was 0.9074, the adjusted *R*^2^ was 0.8657 and the predicted *R*^2^ was 0.7149. The values of all three terms are close to one, which indicates the established model had excellent predictive performance [32]. The adequate precision, which represents the signal-to-noise ratio, is greater than four, implying adequate model discrimination [33]. The *F*-value of the lack of fit is 2.18, which means that lack of fit was not significant by comparison with pure error. 

The final reduced mathematical predicted model in terms of coded factors for the adhesion strength (*Y_M_*), which was obtained after eliminating the insignificant terms, is given as follows: *Y_M_* = 9.88 + 1.22*A −* 0.14*B* − 0.51*C* + 0.18*D* + 0.52*AD*− 0.46*A*^2^ − 0.58*B*^2^ − 0.61*C*^2^ − 0.74*D*^2^(4)
where the final reduced empirical model in terms of actual factors is as follows:*Y_M_* = −8.29124 + 1.42477*A* + 0.25065*B* + 6.32474 × 10^−3^*C* + 0.16199*D* + 0.017188*AD* − 0.20551*A*^2^ − 2.59954 × 10^−3^*B*^2^ − 3.79622 × 10^−6^*C*^2^ − 1.84661 × 10^−3^*D*^2^(5)

In addition, according to the sum of squares of the laser process parameter variables in Table 4, the effect of the laser process parameters on adhesion strength are as follows: laser power (*A*) > scanning speed (*C*) > pulse frequency (*D*) > scanning line interval (*B*).

### 3.2. Effect of Process Parameters on the Responses

Figure 6 is a perturbation plot that illustrates the effect of four key laser process parameters at the center point on the adhesion strength in the design space. It can be clearly seen from the figure that the laser power has a large positive effect on adhesion strength, while scanning speed has a large negative effect on adhesion strength. This is consistent with the previous work reported by Xu et al. [34]. This phenomenon can be explained as follows: The adhesion strength between electroless copper and CFCs largely depends on the surface roughness of the CFC sheet, and a greater roughness results in a higher adhesion strength [35]. The surface roughness is directly related to the laser energy density and the irradiation time. As shown in Figure 3a, the pristine CFC sheet had a very smooth surface. Following nanosecond laser ablation, numerous microcavity structures (Figure 3b) interspersed with irregular granular nanoprotrusions (Figure 3c) emerged on the CFC surface. Additionally, a higher laser power meant that more energy was absorbed by the CFC sheet, leading to the formation of more micro/nanostructures on the surface of the CFC sheet (Figure 3d–f) and an increased adhesion strength between the electroless copper and CFCs. However, a higher scanning speed shortened the laser irradiation time, causing less energy to be input to the CFC sheet and reduced adhesion strength. At the same time, it can be observed from Figure 6 that the scanning line interval (B) had a little effect on the adhesion strength. Simultaneously, the adhesion strength increased with the scanning line interval until it reached its central value and then started to decrease as scanning line interval increased beyond its center limit; this result is consistent with the previous results reported by Qin et al. [36]. It is well known that the laser line-by-line scanning method usually creates a groove structure on the sample [37]. For low scanning line intervals, closer areas of laser line-by-line scanning were generated, resulting in an incomplete groove structure and a reduced surface roughness, thus causing weak adhesion strength. Further increasing the scanning line interval towards the center value, the adhesion strength increased with the roughness of the CFC surface. However, increasing the scanning line interval beyond the central value resulted in a smaller laser ablation area, leading to reduced surface roughness, subsequently decreasing the adhesion strength. In Figure 6, it can also be seen that both too-low or too-high pulse frequency caused low adhesion strength, and the change trend was consistent with the effect of the scanning line interval on adhesion strength.

Figure 7a,b show the interaction effect of laser power and scanning line interval on adhesion strength. It is evident that the adhesion strength between the metal layer and CFCs tended to increase the higher the laser power. This is because the increase in the laser power led to an increase in laser energy density. Then, the CFC sheet absorbed enough laser energy and formed more micro/nanostructures on the CFC surface. Therefore, the adhesion strength increased gradually until the laser power increased to the limit. Moreover, it can be seen from Figure 7a,b that the adhesion strength increased first and then decreased with the increase in the scanning line interval. A lower scanning line interval reduced the roughness of the CFC surface. This may have been due to the cumulative thermal effects resulting from the high overlap percentage of the laser scanning line interval. At the same time, a higher scanning line interval meant that more areas between adjacent scanning lines were not ablated by the laser, so the roughness of CFC surface was naturally lower, and the adhesion strength was poor.

The interaction effect of laser power and scanning speed on adhesion strength is presented in Figure 8a,b. It is clear that the adhesion strength tended to increase for higher laser powers and lower scanning speeds. The increase in laser power and the decrease in scanning speed resulted in the increase in the laser energy density, which effectively improved the roughness of the CFC surface and accordingly increased the adhesion strength of the electroless copper.

The interaction effect of laser power and pulse frequency on adhesion strength is illustrated in Figure 9a,b. The optimal adhesion strength was attained with an appropriate pulse frequency and high laser power. Higher laser power caused more micro/nanostructures to form on the surface of the CFC sheet, which was beneficial for the bond strength.

Figure 10a,b show the interaction effect of scanning line interval and scanning speed on adhesion strength. The optimal scanning line interval and appropriate scanning speed are advantageous for achieving higher adhesion strength. The reasons are the same as those discussed above.

Figure 11a,b present the interaction effect of scanning speed and pulse frequency on adhesion strength. The adhesion strength was relatively low at higher or lower scanning speed and pulse frequency. Thus, the adhesion strength was enhanced in the near-central levels of scanning speed and pulse frequency.

Figure 12a,b show the interaction effect of scanning line interval and pulse frequency on adhesion strength, which is similar to that of scanning speed and pulse frequency on adhesion strength shown in Figure 11.

### 3.3. Validation of the Developed Model

To validate the developed model derived from multiple regression analysis, three groups of optimized laser parameters were chosen randomly within the ranges in Table 1 to conduct confirmation experiments. The actual values of the results in terms of the average of three trials were calculated. Table 5 shows the actual values, predicted results, and calculated percentage error of the confirmation experiments. It shows that the maximum relative error of the prediction for the optimal parametric combination was less than 5.0%, indicating that the developed model can yield near-accurate result. The relationship between the experimental and estimated values of adhesion strength is shown in Figure 13. It can be found that the percentage error between the actual and predicted values was small, indicating that the developed models are adequate, and the predicted data were in good agreement with the measured values. Therefore, it can be concluded that the developed model could successfully predict adhesion strength. In addition, using a numerical optimization method, Design Expert analysis revealed that the optimal laser power, scanning line interval, scanning speed, and laser pulse frequency for adhesion strength were 5.5 W, 48.2 μm, 834.0 mm/s and 69.5 kHz, respectively.

## 4. Conclusions

The effects of laser process parameters on the adhesion strength between electroless copper and CFCs were investigated using RSM. The following conclusions could be drawn from this study:

(1) A four-factor, five-level CCD was successfully employed to develop a mathematical model for optimizing the laser process parameters that affect adhesion strength. The ANOVA results showed that the developed model could be adequately applied to evaluate the adhesion strength at a 95% confidence level.

(2) Laser power has a large positive effect on adhesion strength, which is negatively impacted by scanning speed. The laser pulse frequency and scanning line interval have little effect on adhesion strength.

(3) The maximum relative error of the prediction at the optimal parametric combination was less than 5.0%, indicating that the developed models could adequately predict the results of the adhesion strength within the design space of the laser process parameters presented in this paper. 

(4) Under optimal process conditions (laser power of 5.5 W, scanning line interval of 48.2 μm, scanning speed of 834.0 mm/s, and pulse frequency of 69.5 kHz), electroless copper on CFCs could be produced with an adhesion strength of 10.91 MPa.

## Figures and Tables

**Figure 1 micromachines-14-02168-f001:**
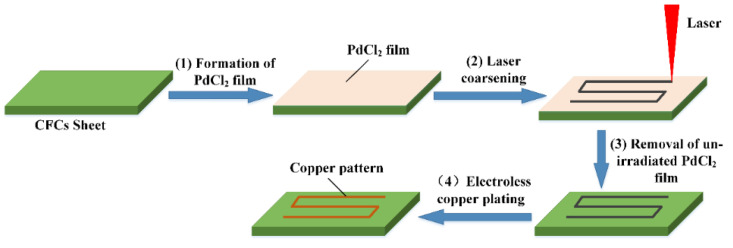
Schematic diagram of the fabrication process of laser-induced copper plating on CFC surface.

**Figure 2 micromachines-14-02168-f002:**
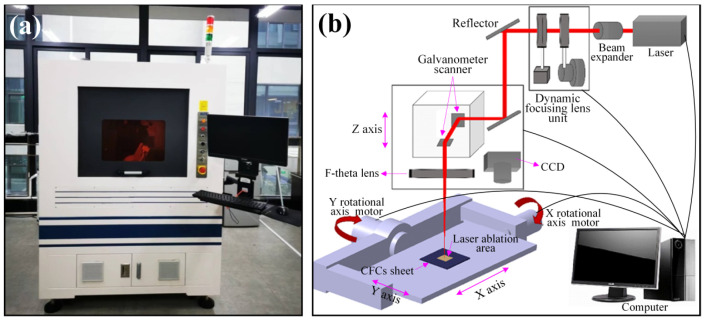
Nanosecond ultraviolet laser 3D process system: (**a**) equipment photograph and (**b**) schematic diagram.

**Figure 3 micromachines-14-02168-f003:**
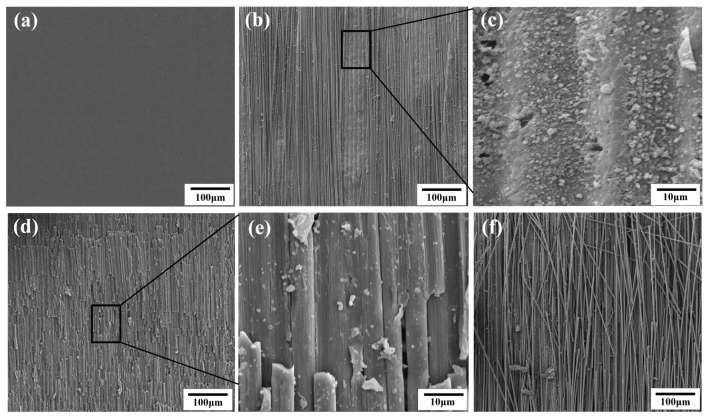
SEM images of pristine CFC surface (**a**) and laser-ablated CFC surface (**b**–**f**); (**c**,**e**) are the local magnifications of (**b**,**d)**, respectively. Laser power for (**b**,**d**,**f**) is 2.5 W, 5 W and 7 W, respectively, while other laser parameters (*U* = 40 μm, *V* = 300 mm/s, *f* = 60 kHz) remained unchanged and were scanned only once with the laser.

**Figure 4 micromachines-14-02168-f004:**
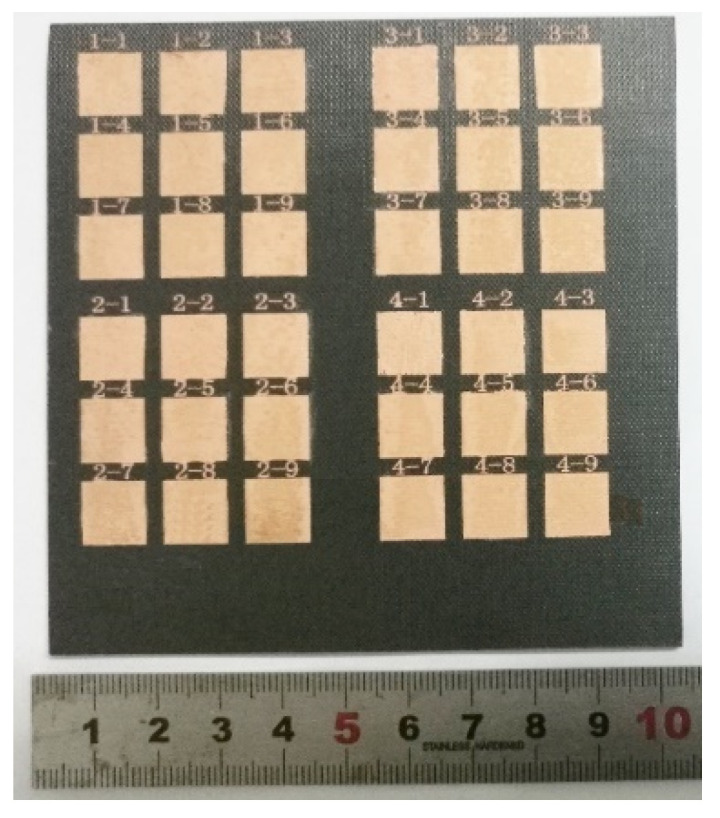
Specimen photograph of electroless copper pattern on CFC surface.

**Figure 5 micromachines-14-02168-f005:**
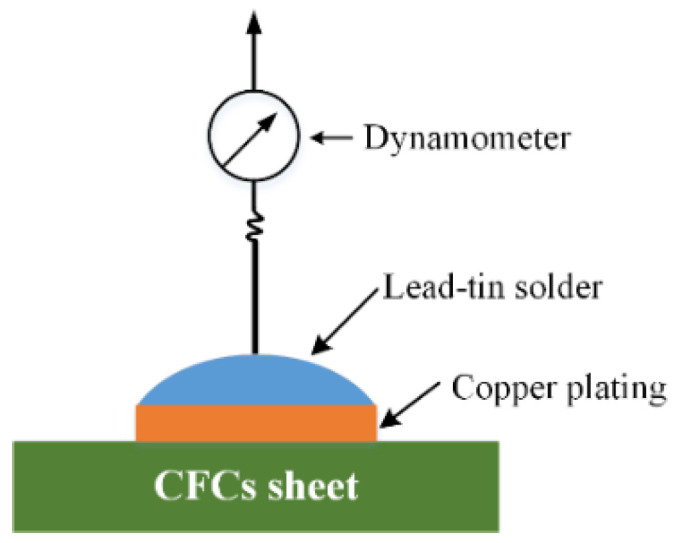
Schematic diagram of vertical pulling force measurement.

**Figure 6 micromachines-14-02168-f006:**
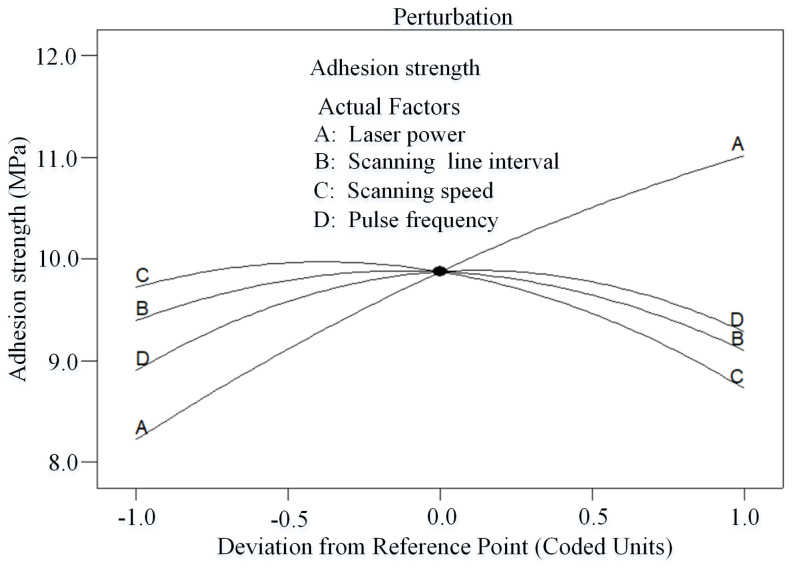
Perturbation plot showing the effect of all factors on the adhesion strength.

**Figure 7 micromachines-14-02168-f007:**
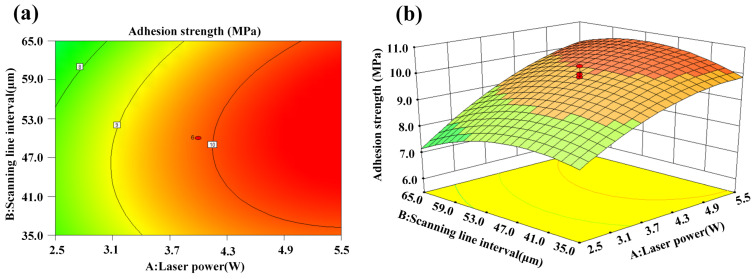
Interaction effect of laser power and scanning line interval on adhesion strength: (**a**) contour plot and (**b**) 3D surface plot.

**Figure 8 micromachines-14-02168-f008:**
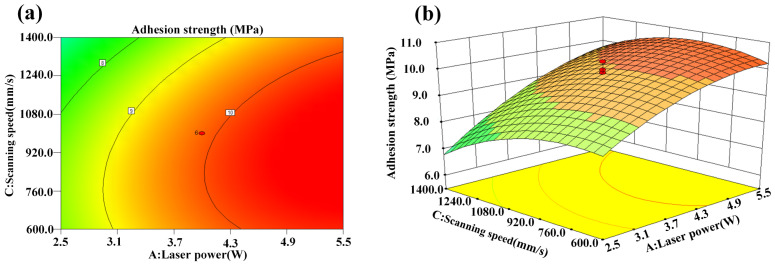
Interaction effect of laser power and scanning speed on adhesion strength: (**a**) contour plot and (**b**) 3D surface plot.

**Figure 9 micromachines-14-02168-f009:**
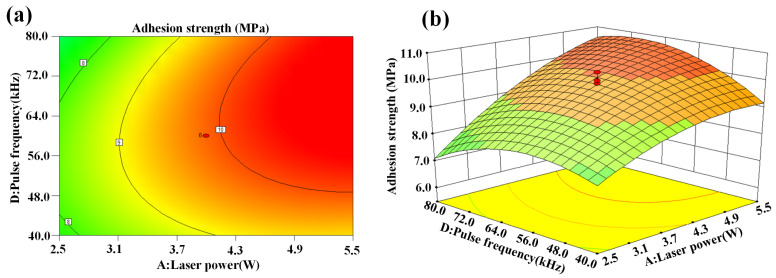
Interaction effect of laser power and pulse frequency on adhesion strength: (**a**) contour plot and (**b**) 3D surface plot.

**Figure 10 micromachines-14-02168-f010:**
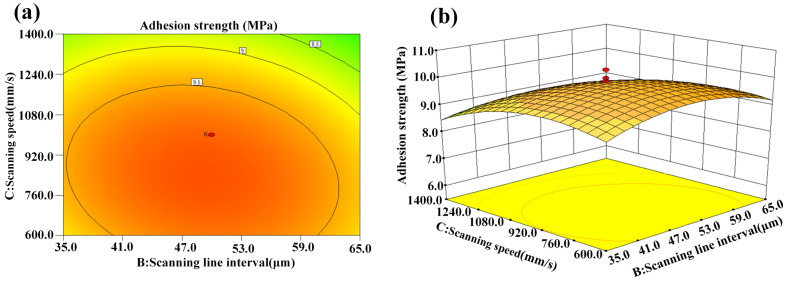
Interaction effect of scanning line interval and scanning speed on adhesion strength: (**a**) contour plot and (**b**) 3D surface plot.

**Figure 11 micromachines-14-02168-f011:**
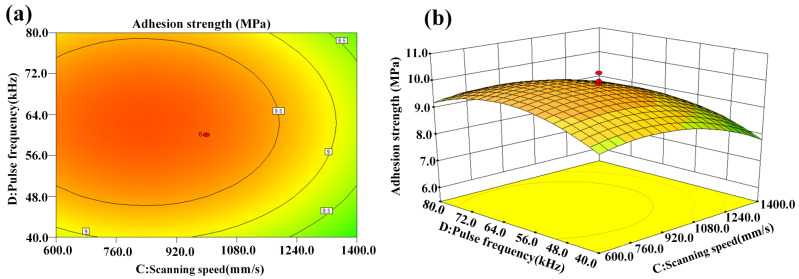
Interaction effect of scanning speed and pulse frequency on adhesion strength: (**a**) contour plot and (**b**) 3D surface plot.

**Figure 12 micromachines-14-02168-f012:**
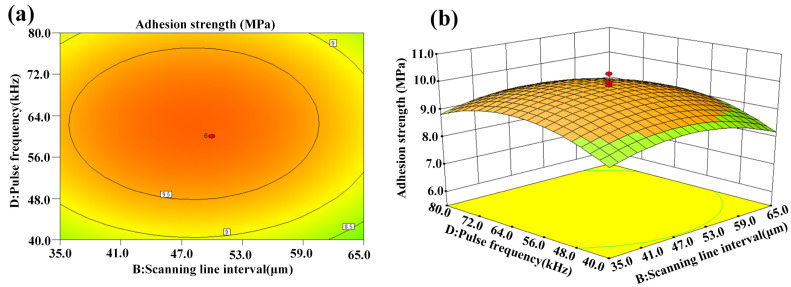
Interaction effect of scanning line interval and pulse frequency on adhesion strength: (**a**) contour plot and (**b**) 3D surface plot.

**Figure 13 micromachines-14-02168-f013:**
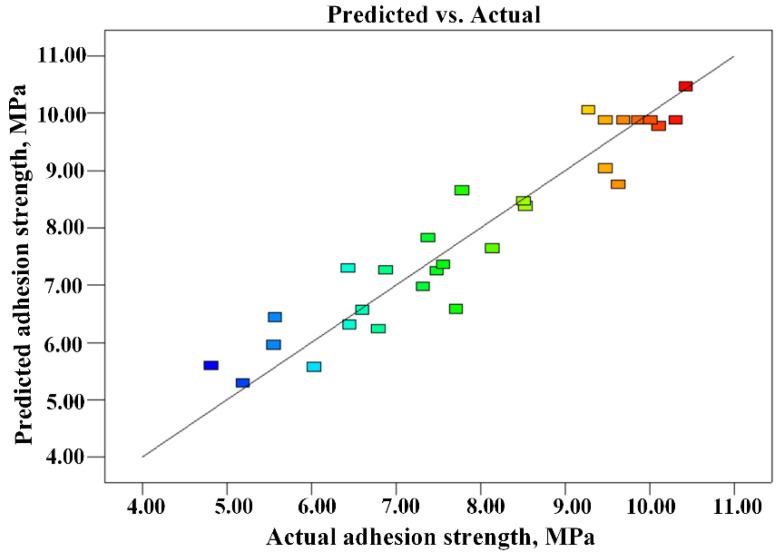
Plots of the actual vs. predicted on adhesion strength.

**Table 1 micromachines-14-02168-t001:** Laser process parameters and their levels used in RSM design.

Parameter	Unit	Notation	Level				
			−2	−1	0	1	2
Laser power	W	*A*	1.0	2.5	4.0	5.5	7.0
Scanning line interval	μm	*B*	20	35	50	65	80
Scanning speed	mm/s	*C*	200	600	1000	1400	1800
Pulse frequency	kHz	*D*	20	40	60	80	100

**Table 2 micromachines-14-02168-t002:** Design matrix and measured responses.

Std Order	Run Order	Laser Process Parameter				Adhesion Strength *Y_M_* (MPa)
		*A* (W)	*B* (μm)	*C* (mm/s)	*D* (kHz)	
1	21	2.5	35	600	40	7.48
2	8	5.5	35	600	40	7.78
3	18	2.5	65	600	40	7.32
4	4	5.5	65	600	40	8.53
5	7	2.5	35	1400	40	6.79
6	28	5.5	35	1400	40	8.14
7	15	2.5	65	1400	40	5.55
8	16	5.5	65	1400	40	7.56
9	11	2.5	35	600	80	7.71
10	1	5.5	35	600	80	9.28
11	19	2.5	65	600	80	6.45
12	2	5.5	65	600	80	10.11
13	24	2.5	35	1400	80	6.03
14	30	5.5	35	1400	80	9.48
15	22	2.5	65	1400	80	5.19
16	6	5.5	65	1400	80	9.63
17	20	1.0	50	1000	60	4.81
18	25	7.0	50	1000	60	10.43
19	26	4.0	20	1000	60	7.38
20	27	4.0	80	1000	60	6.88
21	9	4.0	50	200	60	8.51
22	10	4.0	50	1800	60	5.57
23	29	4.0	50	1000	20	6.60
24	3	4.0	50	1000	100	6.43
25	12	4.0	50	1000	60	9.91
26	23	4.0	50	1000	60	9.87
27	5	4.0	50	1000	60	10.01
28	14	4.0	50	1000	60	9.69
29	13	4.0	50	1000	60	10.31
30	17	4.0	50	1000	60	9.48

**Table 3 micromachines-14-02168-t003:** ANOVA for the adhesion strength model before stepwise elimination.

Source	Sum of Squares	*df*	Mean Square	*F*-Value	*p*-Value Prob > *F*	
Model	79.09	14	5.65	18.06	<0.0001	significant
*A*	35.60	1	35.60	113.84	<0.0001	
*B*	0.47	1	0.47	1.50	0.2403	
*C*	6.17	1	6.17	19.73	0.0005	
*D*	0.80	1	0.80	2.57	0.1299	
*AB*	1.35	1	1.35	4.32	0.0621	
*AC*	1.27	1	1.27	4.07	0.1116	
*AD*	4.25	1	4.25	13.60	0.0022	
*BC*	0.45	1	0.45	1.42	0.2512	
*BD*	7.563 × 10^−4^	1	7.563 × 10^−4^	2.418 × 10^−3^	0.9614	
*CD*	1.406 × 10^−3^	1	1.406 × 10^−3^	4.497 × 10^−4^	0.9474	
*A* ^2^	5.86	1	5.86	18.75	0.0006	
*B* ^2^	9.38	1	9.38	30.00	<0.0001	
*C* ^2^	10.12	1	10.12	32.36	<0.0001	
*D* ^2^	14.96	1	14.96	47.85	<0.0001	
Residual	4.69	15	0.31			
Lack of fit	4.29	10	0.43	2.45	0.2100	Not significant
Pure error	0.40	5	0.080			
Core total	83.78	29				
Standard deviation = 0.56					*R*^2^ = 0.9440	
Mean = 7.96					Adjusted *R*^2^ = 0.8918	
Coefficient of variation = 7.02					Predicted *R*^2^ = 0.6981	
Predicted residual error of sum of squares (PRESS) = 25.30					Adequate precision = 14.968	

**Table 4 micromachines-14-02168-t004:** ANOVA for the adhesion strength model after stepwise elimination.

Source	Sum of Squares	*df*	Mean Square	*F*-Value	*p*-Value Prob > *F*	
Model	76.02	9	8.45	21.77	<0.0001	Significant
*A*	35.60	1	35.60	91.74	<0.0001	
*B*	0.47	1	0.47	1.20	0.2854	
*C*	6.17	1	6.17	15.90	0.0007	
*D*	0.80	1	0.80	2.07	0.1658	
*AD*	4.25	1	4.25	10.96	0.0035	
*A* ^2^	5.86	1	5.86	15.11	0.0009	
*B* ^2^	9.38	1	9.38	24.18	<0.0001	
*C* ^2^	10.12	1	10.12	26.08	<0.0001	
*D* ^2^	14.96	1	14.96	38.56	<0.0001	
Residual	7.76	20	0.39			
Lack of fit	7.36	15	0.49	2.18	0.2273	Not significant
Pure error	0.40	5	0.080			
Core total	83.78	29				
Standard deviation = 0.62					*R*^2^ = 0.9074	
Mean = 7.96					Adjusted *R*^2^ = 0.8657	
Coefficient of variation = 7.82					Predicted *R*^2^ = 0.7149	
Predicted residual error of sum of squares (PRESS) = 23.89					Adequate precision = 14.394	

**Table 5 micromachines-14-02168-t005:** Prediction and validation test results.

Exp. No.	*A* (W)	*B* (μm)	*C* (mm/s)	*D* (kHz)	Adhesion Strength (MPa)		|Error| (%)
					Actual	Predicted	
1	2	40	500	60	6.63	6.95	4.82
2	6	70	1500	40	6.05	6.26	3.47
3	5	30	1000	70	9.26	9.71	4.86

## Data Availability

The data that support the findings of this study are available from the corresponding author, Zhongli Qin, upon reasonable request.

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
