# Peer review of "The Influence of Laser Process Parameters on the Adhesion Strength between Electroless Copper and Carbon Fiber Composites Determined Using Response Surface Methodology"

_micromachines, 2023, doi:10.3390/mi14122168_

Round 1

Reviewer 1 Report

Comments and Suggestions for Authors

1.    Line 14 and 16: If the abbreviation, CFC is defined, it should be used.

2.    Line 19 and 20: If the abbreviations are not reused, they should not be defined.

3.    Line 66: “discuss” should be “discussion”.

4.    In the introduction, the objective is not clear. The purpose of study should be concretely described.

5.    Line 100, 102, 110, 112, 118, and 122: The numbers should be subscripted.

6.    The more detailed schematic should be presented in Fig. 2(b).

7.    The pulse width, beam size, shape, profile, and M2 should be presented in the experiment.

8.    The role of PdCl2 should be described in the experiment.

9.    Line 159: M should be subscripted.

10.  Line 160: f was used as a pulse frequency in the line 89.

11.  Line 145 and 169: Indent should be removed.

12.  Line 179: CCD was already defined in the line 86.

13.  Line 195: ANOVA was already defined in the line 90.

14.  Line 236: “this” should be “This”.

15. In fig. 6, I wonder how the data described in the plot is related to adhesion strength.

16. More explanations should be provided in Figs. 7-12.

Comments on the Quality of English Language

This manuscript should be corrected by a native speaker for technical writing.

Author Response

Thank you very much for your useful comments and suggestions on the content of our manuscript. We have modified the manuscript accordingly and highlighted all changes using red font in the revised manuscript. Detailed corrections are listed below point by point. Please see the attachment.

Reviewer 2 Report

Comments and Suggestions for Authors

In this paper, a nanosecond ultraviolet laser is used to electroless copper plating on the surface of carbon fiber composites. Based on the response surface method (RSM), four key process parameters were optimized and the optimum process conditions were found. The central combination design (CCD) was used for experimental design, and the analysis of variance (ANOVA) was used for measurement. The adequacy and significance of the developed regression model were evaluated. However, several issues need to be identified:

1.     At present, it is not clear whether the calculation method of adhesive force YM in formula (1) in this work is correct, reliable, and reasonable. Therefore, it should be explained whether there are suitable theories and literature as support.

2.     In Table 3 and Table 4, symbols of the interaction effect of various parameters such as PU, PV, PF, UV and Vf are not explained in the paper. Please write in detail in the text.

3.     In Table 3, the P-value of the interaction between power and Scanning line interval (PU) is 0.0621, which is very close to the 0.05 mentioned in the paper. Whether there is an error problem and whether it is verified again to prevent missing the influence of this factor.

4.     It is mentioned on page 9, lines 240-242: “A higher laser power means that more energy is absorbed by CFCs sheet, thus leads to the formation of more micro/nanostructures on the surface of CFCs sheet (Fig.3) and an increased adhesion 242 strength. "However, Figure 3 only shows the SEM scan of the laser power at 5W, whether this is enough to draw a conclusion to support the stated view, and whether there are more data to confirm this view.

5.     In Table 5, the authors propose: “To validate the developed model derived from multiple regression analysis, three groups of optimized laser parameters were chosen randomly within the ranges of the Table 1 to conduct confirmation experiments. "But the range of parameters taken in Table 5 is much smaller than that given in Table 1, and it has only been verified by three experiments." It seems that the number of experimental verifications is too small, so whether the results obtained are reasonable and rigorous.

7.     The authors do not indicate the limitations of the study, the future research plan, and the direction of the following research in this paper.

Comments on the Quality of English Language

6.     Most of the tables in the article have different pages, can you make the layout more reasonable and try to put it on one page?

Author Response

(The authors gave the same response as above.)

Round 2

Reviewer 1 Report

Comments and Suggestions for Authors

All the questions are adequately addressed. 

Line 209 "AVONA" seems to be "ANOVA".

What is the Mfor the Gaussian beam?

Comments on the Quality of English Language

Some typos should be corrected.

Author Response

Thank you very much for such a timely reviewer comment. In order to facilitate the distinction with the last paper revision, we have modified the manuscript accordingly and highlighted all changes using green background in the revised manuscript. Detailed corrections are listed below point by point. Please see the attachment. 
